# Harris Hawks Sparse Auto-Encoder Networks for Automatic Speech Recognition System

**Mohammed Hasan Ali [1], Mustafa Musa Jaber [2], Sura Khalil Abd [2], Amjad Rehman [3], Mazhar Javed Awan [4,\*], Daiva Vitkutė-Adžgauskienė [5], Robertas Damaševičius [5,\*] and Saeed Ali Bahaj [6]**

1 Computer Techniques Engineering Department, Faculty of Information Technology, Imam Ja'afar Al-sadiq University, Baghdad 10021, Iraq; mh180250@gmail.com
2 Department of Computer Science, Dijlah University College, Baghdad 00964, Iraq; mustafa.musa@duc.edu.iq (M.M.J.); sura.khalil@duc.edu.iq (S.K.A.)
3 Artificial Intelligence and Data Analytics Laboratory, College of Computer and Information Sciences (CCIS), Prince Sultan University, Riyadh 11586, Saudi Arabia; rkamjad@gmail.com
4 Department of Software Engineering, University of Management and Technology, Lahore 54770, Pakistan
5 Department of Applied Informatics, Vytautas Magnus University, 44404 Kaunas, Lithuania; daiva.vitkute-adzgauskiene@vdu.lt
6 MIS Department, College of Business Administration, Prince Sattam bin Abdulaziz University, Alkharj 11942, Saudi Arabia; s.bahaj@psau.edu.sa
* Correspondence: mazhar.awan@umt.edu.pk (M.J.A.); robertas.damasevicius@vdu.lt (R.D.)

**Abstract:** Automatic speech recognition (ASR) is an effective technique that can convert human speech into text format or computer actions. ASR systems are widely used in smart appliances, smart homes, and biometric systems. Signal processing and machine learning techniques are incorporated to recognize speech. However, traditional systems have low performance due to a noisy environment. In addition to this, accents and local differences negatively affect the ASR system's performance while analyzing speech signals. A precise speech recognition system was developed to improve the system performance to overcome these issues. This paper uses speech information from jim-schwoebel voice datasets processed by Mel-frequency cepstral coefficients (MFCCs). The MFCC algorithm extracts the valuable features that are used to recognize speech. Here, a sparse auto-encoder (SAE) neural network is used to classify the model, and the hidden Markov model (HMM) is used to decide on the speech recognition. The network performance is optimized by applying the Harris Hawks optimization (HHO) algorithm to fine-tune the network parameter. The fine-tuned network can effectively recognize speech in a noisy environment.

**Keywords:** automatic speech recognition; Mel-frequency cepstral coefficients; sparse auto-encoder neural network; hidden Markov model; natural language processing; speech recognition

## 1. Introduction

Artificial intelligence (AI) methods [1] evolve rapidly and are increasingly creating effective communication systems. AI can both effectively analyze and recreate the human voice, and automatic speech recognition (ASR) systems [2] have been created to achieve communication and dialogue like real people's conversation. The ASR system combines the fields of linguistics, computer science, natural language processing (NLP), and computer engineering. The system needs a training process to understand the individual speakers and recognize the speeches; here, speakers read the text and vocabularies to get the speaker's inner details (speaker-dependent). Most of the ASR system does not require the speaker-independent system's training process. Advancement of machine and deep learning techniques is highly involved in ASR to improve the Persian speech classification in an efficient [3]. However, ASR has been affected negatively by loud and noisy environmental factors fuzzy phoneme [4], which create challenging issues and causes for ambiguous ASR.

Voice-powered user interfaces (VUIs) [5] are integrated into ASR to solve the issue of loud and noisy environments. The VUIs and voice assistants allow users to speak to the machine and the machine converts the speech into actions. However, false interpretations and imprecisions create further complexity with vision based sign language [6]. Misinterpretations occur due to the understanding of sentences, words, and their relationship with human aspects. The semantic sentence understanding is also one of the main reasons for the inaccurate ASR results. Then, machine learning methods like the hidden Markov model (HMM), the support vector machine (SVM), deep neural networks (DNN), etc. [7–9] are employed to overcome the above difficulties. However, the developed ASR system has time and lack of efficiency issues. The recognition system provides for variations of voices, which requires continuous learning and training procedures.

The ASR system has minimum accuracy when it receives information from speech based on loud and background noise [10,11]. Noise is important for malware attacks in Image-based classification [12]. Especially in offices, public spaces, and urban outdoors, noise will be one of the significant challenges in speech recognition [13]. Noise occurred due to variations in locales and the accents of speaker's speech. Noise can be reduced by incorporating headphones and particular microphones. Therefore, additional computational cost and complexity are needed when using additional devices that are not desirable in the ASR system. So, the automatic ASR system has been developed by using machine learning or deep learning algorithms [14,15] such as recurrent neural networks [16] and the encoder–decoder with an attention mechanism [17]. Speech enhancement techniques are important for maximizing the accuracy in DDos attacks in real time [18]. This system works in conjunction with signal processing, which examines each frequency and the respective speech modulations [19]. The derived information is processed using machine learning techniques that extract meaningful patterns. Finally, the system is decomposed into the training and testing phase. In the training stage, different features are utilized to train the data to recognize the exact speech obtained from various environments. The training process observes every change, variation, and speech modulation that reduces the difficulties in the noisy environment speech recognition process. Then, the deviation between the exact spoken speech and the predicted speech is minimized by network parameter analysis. Continuous examination of the network parameters and the fine-tuning process help to reduce the maximum error rate problem. Minimum-error problems directly improve the ASR system's precision and maintain the system flexibility and robustness of the system.

The Mel-frequency cepstral coefficients (MFCCs) approach [20,21] is applied to derive various features from the collected speech signal. The training and testing process is then initiated via the sparse autoencoder algorithm, which improves the overall recognition accuracy. Recognition decision is handled through the hidden Markov model (HMM) [19] and the network parameter fine-tuning process.

Thus, the main contribution of the proposed system is reducing the maximum error rate problem and improving the precision of the ASR system.

The rest of the paper is organized as follows. Section 2 analyzes the various researchers' opinions regarding the speech recognition process. Section 3 discusses the introduced metaheuristic algorithm-based ASR process and the system effectiveness, which is evaluated in Section 4. Finally, the summarization of the entire work is discussed in Section 5.

## 2. Related Work

Recently, using different deep learning approaches, tremendous advances have been achieved in the field of automated voice recognition (ASR) [22–24]. In this section, a complete comparison of cutting-edge strategies currently being employed with a specific emphasis on the various deep learning methods. Lokesh et al. [25] proposed a bidirectional recurrent neural network (BRNN) with a self-organizing map (SOM)-based classification scheme for Tamil speech recognition. To begin, the input voice signal is preprocessed using the Savitzky–Golay filter to remove background noise and improve the signal. Perceptual linear predictive coefficients were split to improve the accuracy of the classification. The

feature vector is shifted in measure and SOM is used to select the appropriate length of the feature vector. Finally, the Tamil numerals and words are arranged using a BRNN classifier using the fixed-length feature vector from SOM as input, known as BRNN-SOM. Ismail et al. [26] aimed to develop speech recognition systems and improve the interaction between the home appliance and the human by giving voice commands. Speech signals are processed by dynamic time warping (DTW) techniques and use SVM to recognize the voice with up to 97% accuracy.

Hori et al. [14] used deep convolution encoder and long-short-term memory (LSTM) recurrent neural networks (RNN) to recognize end-to-end speech. This process uses the connectionist temporal classification procedure while investigating the audio signals. The convolution network uses the VGG neural network architecture, which works jointly with the encoder to investigate the speech signal. Then, the memory network stores every speech signal, which improves the system performance compared to existing methods. Finally, the framework introduced is applied to the Chinese and Japanese datasets, and the system ensures a 5% to 10% error rate.

Neamah et al. [15] recommend continual learning algorithms such as the hidden Markov model and deep learning algorithms to perform automatic speech recognition. Here, a deep learning network learns the speech features derived from the Mel-frequency coefficient approach. The learning process minimized the deviation between the original audio and the predicted audio. The trained features are further evaluated using the Markov model to improve offline mode's overall recognition accuracy.

Khan et al. [27] selected a time-delayed neural network to reduce the problem of limited language analysis using the Hindi speech recognition system. The Hindi speech information is collected from Mumbai people that are processed using an i-vector adapted network. The network considers time factors when investigating speech characteristics. This process reduces training time because the delay network maintains all processed speech information. Furthermore, the effective utilization of the network parameters increases the recognition accuracy up to 89.9%, which is a 4% average improvement compared to the existing methods.

Mao et al. [28] created a multispeaker diarization model to recognize long conversation-based speech. The method uses audio–lexical interdependency factors to learn the model for improving the word diarization process. This learning process generates a separate training setup for the diarization and ASR systems. The training setup helps identify long conversation speech with minimum effort because the data augmentation and decoding algorithm recognizes the speech accurately.

Kawase et al. [18] suggested a speech enhancement parameter with a genetic algorithm to create the automatic speech recognition system. This study aims to improve the recognition accuracy while investigating the noisy speech signal. Here, a genetic algorithm is applied to investigate the speech parameter and the noise features are removed from the audio, which helps improve the overall ASR system.

Another stream of research on ASRs focused on speech emotion recognition (SER) [29]. In the context of human–computer or human–human interaction applications, the challenge of identifying emotions in human speech signals is critical and extremely difficult [30]. The blockchain based IoT devices and systems have been created [31]. For example, Khalil et al. [32] reviewed deep learning techniques to examine emotions from the speech signal. This paper will examine deep learning techniques, functions, and features to extract human emotions from audio signals. This analysis helps to improve the speech recognition process further. Fahad et al. [33] created a deep learning with a hidden Markov model-based speech recognition system using the epoch and MFCC features. First, the speech features are derived by computing the maximum likelihood regression value. Then, the derived features are processed by the testing and training phase to improve the overall prediction of speech emotions. The effectiveness of the system was measured using information from the emotional dataset of the Interactive Emotional Dyadic Motion Capture (IEMOCAP), and the system ensures high results up to $\pm 7.13\%$ compared to existing methods.

Zhao et al. [34] created a merged convolutional neural network (CNN) with two branches, one one-dimensional (1D) CNN branch and another two-dimensional (2D) CNN branch to learn high-level features from raw audio samples. First, a 1D CNN and a 2D CNN architecture were created and assessed; after the second dense layers were removed, the two CNN designs were fused. Transfer learning was used in the training to speed up the training of the combined CNN. First, the 1D and 2D CNNs were trained. The learnt characteristics of the 1D and 2D CNNs were then reused and transferred to the combined CNN. Finally, the initialization of the merged deep CNN with transferred features was fine-tuned. Two hyperparameters of the developed architectures were chosen using Bayesian optimization in the training. Experiments on two benchmark datasets demonstrate that merged deep CNN may increase emotion classification performance. In another paper, Zhao et al. [35] proposed learning local and global emotion-related characteristics from speech and log-Mel spectrograms using two CNN and LSTM models. The architectures of the two networks are identical, with four local feature learning blocks (LFLBs) and one LSTM layer each. The LFLB, which consists mostly of one convolutional layer and one maximum-pooling layer, is designed to learn local correlations and extract hierarchical correlations. The LSTM layer is used to learn long-term dependencies from locally learnt characteristics. The developed models use the strengths of both networks while overcoming their drawbacks.

Finally, speech recognition methods have been extensively used for medical purposes and disease diagnostics, such as developing biosignal sensors to help people with disabilities speak [36] and fake news to manage sentiments [37]. The audio challenges [38] were captured using two microphone channels from an acoustic cardioid and a smartphone, allowing the performance of different types of microphones to be evaluated. Polap et al. [39] suggested a paradigm for speech processing based on a decision support system that can be used in a variety of applications in which voice samples can be analyzed. The proposed method is based on an examination of the speech signal using an intelligent technique in which the signal is processed by the built mathematical transform in collaboration with a bioinspired heuristic algorithm and a spiking neural network to analyze voice impairments. Mohammed et al. [40] adopted a pretrained CNN for recognition of speech pathology and explored a distinctive training approach paired with multiple training methods to expand the application of the suggested system to a wide variety of vocal disorders-related difficulties. The suggested system has been evaluated using the Saarbrücken Voice Database (SVD) for speech pathology identification, achieving an accuracy of 95.41%. Lauraitis et al. in [41,42] developed a mobile application that can record and extract pitch contour features, MFCC, gammatone cepstral coefficients, Gabor (analytic Morlet) wavelets, and auditory spectrograms for speech analysis and recognition of speech impairments due to the early stage of central nervous system disorders (CNSD) with up to 96.3% accuracy. The technology can be used for automated CNSD patient health status monitoring and clinical decision support systems, and a part of the Internet of Medical Things (IoMT).

In summary, speech recognition played a vital role in different applications. Therefore, several intelligent techniques are incorporated to improve speech recognition effectiveness. However, in the loud and noisy environment, speech signals are difficult to recognize accurately. Therefore, metaheuristics-optimized techniques, specifically the Harris Hawk (HH) heuristic optimization algorithm [43], are incorporated with the traditional machine learning techniques to improve the overall recognition accuracy. HH has been successfully used before for various other applications such as feature selection [44], big data-based techniques using spark [45–49], pronunciation technology [50,51] and image chain based optimizers thresholding [52,53], and deep learning [54,55]. However, traditional systems have computational complexity due to a noisy environment. In addition to this, accents and local differences affect the performance of the ASR system. This causes the system reliability and flexibility to be affected while analyzing speech signals.

The detailed working process of the introduced ASR system is discussed in Section 3.

## 3. Methodology

### 3.1. Data Set Description

This section examines the effectiveness of the proposed Harris Hawks sparse auto-encoder networks (HHSAE-ASR) framework. The jim-schwoebel voice datasets applied on our experiments [56]. The dataset consists of several voice datasets that are widely used to investigate the effectiveness of the introduced system.

### 3.2. Harris Hawks Sparse Auto-Encoder Networks (HHSAE)-ASR Framework

This system aims to reduce the computation complexity while investigating the loud and noisy environment speech signal. The HHSAE-ASR framework utilizes the learning concepts that continuously train the system using speech patterns. Then, metaheuristic techniques, specifically the Harris Hawks (HH) algorithm, are applied to the encoder network to fine-tune the network parameters that minimize the error-rate classification problem. Here, the HH algorithm allows for recognizing the sequence of speech patterns, learning concepts, and the network parameter updating process, and improves the precise rate, robustness, and reliability of the ASR. The HHSAE-ASR framework is then illustrated in Figure 1.

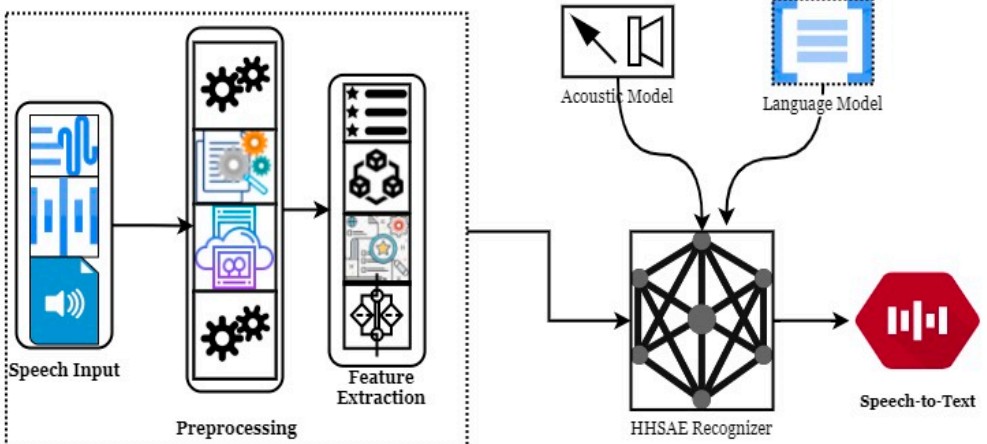

**Figure 1.** Outline of HHASE-ASR framework that includes speech input, speech preprocessing, feature extraction, speech recognition, and speech-to-text modules.

The working process illustrated in Figure 1 consists of several stages, such as the collection of speech signals, preprocessing, feature extraction, and the recognizer. The collected speech signals generated a lot of noisy and inconsistent information that completely affects the quality and precision of the ASR system. Therefore, modulations and changes should be suspected at all frequencies, and irrelevant details should be eliminated.

#### 3.2.1. Speech Signal Preprocessing and Denoising

Here, the spectral deduction approach is applied to the collected speech signal to purify the signal. The method effectively apprises the spectrums in the most straightforward and easiest ways. The spectrum is not affected by time due to the additive noise. For every speech signal $s(n)$, it has a clean signal $cs(n)$ and an additive noise signal $ad(n)$. Therefore, the original speech signal is written as Equation (1).

$$s(n) = cs(n) + ad(n) \tag{1}$$

The clean signal $cs(n)$ is obtained by applying the discrete Fourier transform with the imaginary and the real part, which gives the noise-free speech output signal. The Fourier transform representation of the signal is defined in Equation (2).

$$s(w) = cs(w) + ad(w) \tag{2}$$

$$s(w) = s|(w)|e^{j\varnothing s} \tag{3}$$

The Fourier transform of signal $s(n)$ is obtained by computing the spectrum magnitude $s|(w)|$ and the $\varnothing$ phase spectra value of the noise signal is obtained using Equation (4).

$$ad(w) = |ad[w]|e^{j\varnothing \mathfrak{Y}} \tag{4}$$

The value of the computed noise spectrum value $|ad[w]|$ is more helpful to identify the noisy information of the original speech signal. This noise continuously occurs in a loud and noisy environment, which completely affects the originality of the speech. Therefore, the noise value in $s(w)$ should be replaced by the average noisy spectrum value. This average value is computed from the details of nonspeech activities (speech pause) and speech ineligibility (s) because it does not affect the speech quality. Therefore, the noise-free signal is computed as:

$$cs_e[w] = [|s(w)| - |ad_e[w]|]e^{j\varnothing \mathfrak{Y}} \tag{5}$$

The clean signal $cs_e[w]$ is estimated from the computation of the signal spectrum magnitude $s|(w)|$ of the phase spectrum value and the average noise spectrum value of the noise signal $|ad_e[w]|$. The spectral magnitude is computed to clean the recorded speech signal.

Extraction of features is used to train a Markov model-based convolution network for resolving noisy and loud voice signals. According to the hawk's prey finding behavior, the network's parameters are fine-tuned and updated during this process. The system's robustness and availability are maintained by reducing the number of misclassification errors.

$$cs_e[w] = [|s(w)| - |ad_e[w]|] \tag{6}$$

Then, Equation (6) is applied to identify the power spectrum of the speech signal $s(w)$; $cs_e[w]^2 = |s(w)|^2 - |ad_e[w]|^2$ to estimate the original noise-free signal. The computed spectral values cut off the noise information from the original signal $s(n)$. Then, the inverse Fourier transform is applied on the signal magnitude $|cs_e[w]|$ and the power spectrum $\left|cs_e[w]^2\right|$ to identify the noise-free speech signal.

$$cs_e[\omega]p = |s(\omega)|p - |ad_e[\omega]|p \tag{7}$$

The noise-free signal is computed from the spectral subtraction of the power exponent $p$. Here, the noise signal spectrum deduction is performed according to $p$. If the $p$ has a value of 1, then magnitude is affected by noise and that part is deducted from the signal. If the value of $p$ is 2, the power spectral deduction is applied to obtain the original noise-free signal. Then, the noise removal of the speech signal is summarized in Figure 2.

### 3.2.2. Signal Decomposition and Feature Extraction

The extracted features are more useful to get the important details that improve the overall ASR systems more precisely. The feature extraction process helps maintain the robustness of the ASR system because it helps to investigate the signal $s(n)$ in different aspects. The speech signal $s(n) = cs(n) + ad(n)$ has the length of $N$. Once the noise has been eliminated, $cs(n)$ has been divided based on the trend and fluctuations by applying the wavelet transform. Here, level 4 Daubechies wavelets are utilized to extract five wavelets, such as db14, db12, db10, db8, and db2.

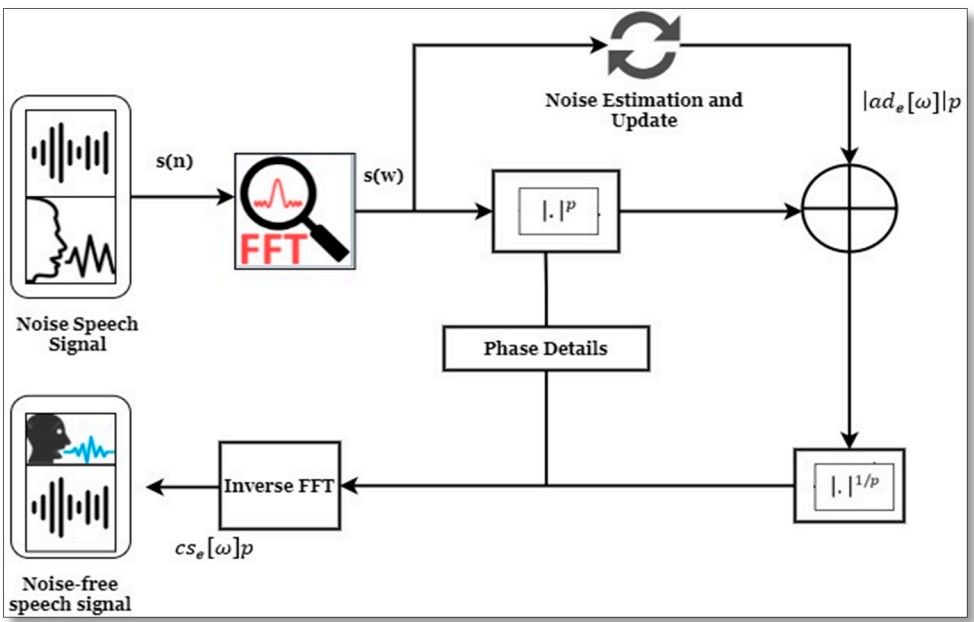

**Figure 2.** Speech signal noise removal using spectral deduction method.

Then, the level of the speech signal mapping process is illustrated as follows.

In level 1, the speech signal $c(n)$ is divided into the first level according to the signal length $N/2$ of the trend $I_1$ and fluctuations $f_1$.

$$c(n) \rightarrow (I_1|f_1) \tag{8}$$

In level 2, the speech signal is divided by $N/4$ length and is obtained from trend $I_1$ and fluctuations $f_1$, which is defined as $I_1 \rightarrow (I_2|f_2)$.

$$c(n) \rightarrow (I_2| \ f_2|f_1) \tag{9}$$

In level 3, the signal is calculated by dividing the $I_2$ and $f_2$ signals that are defined as $I_2 \rightarrow (I_3|f_3)$. Here, the decomposition process is carried out of length $N/8$.

$$c(n) \rightarrow (I_3| \ f_3|f_2|f_1) \tag{10}$$

In level 4, the decomposition is carried out for $N/16$ length, and it is obtained by the $I_3$ and $f_3$ signals that are represented as $I_3 \rightarrow (I_4|f_4)$.

$$c(n) \rightarrow (I_4|f_4| \ f_3|f_2|f_1) \tag{11}$$

According to the above wavelet process, 20 subsignals are obtained according to trends and fluctuations. After that, the signal entropy value ($\mathfrak{ev}$) is estimated, which helps to determine the information of the signal presented in the decomposed signals. The entropy value was obtained according to Equation (12).

$$(\mathfrak{ev})(Q) = \sum_{i=1}^{n} p(q_i) \log p(q_i) \tag{12}$$

The entropy value ($\mathfrak{ev}$) is computed from the random phenomenon of speech signal $Q\{q_1, q_2, \ldots, q_n\}$ and the probability value of $p(q_i)$ of $Q$. Then, according to $Q$, every subsignal entropy value is estimated using Equation (13).

$$I_4 \rightarrow \{Ie_{4k}\}, \ f_j \rightarrow \left\{fe_{jk}\right\} \tag{13}$$

The subsignal entropy value ($\mathfrak{ev}$) is computed from m number of frames, $k = 1, 2, \ldots m$. $j = 1, 2, 3$. According to Equation 13, the entropy values are $Ie_{4k}$ and $fe_{jk}$. These extracted frame entropy values characterize the speech based on emotions because the fluctuations are varying when compared to the normal speaker emotion level. Then, Mel-frequency coefficient features are derived from identifying the characteristics of the speech signal.

$$Mel(f) = 2595 \log\left(1 + \frac{f}{700}\right) \tag{14}$$

The $Mel(f)$ value is obtained from the frequency value of every subsignal derived from the discrete wavelet transform process. The extracted features are trained and learned by the encoder convolution networks to train the feature to perform in any situation. The process of feature extraction is summarized in Figure 3.

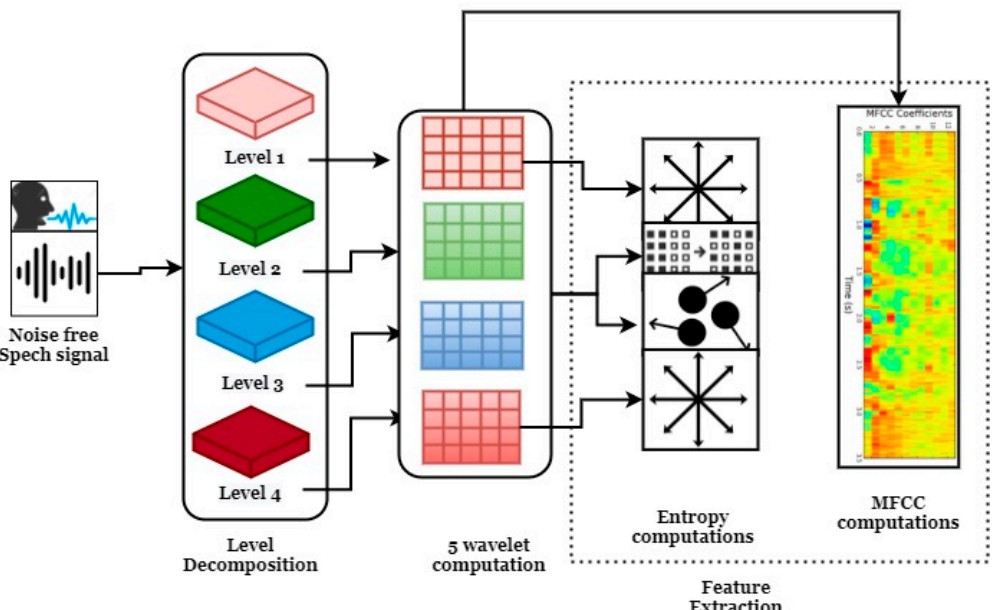

**Figure 3.** Feature extraction using level decomposition, wavelet computation, entropy calculation, and MFCC coefficient computation.

### 3.2.3. Speech Recognition

The convolution network trains the extracted features to recognize the speech signal in different noisy and loud environments. The learning process is done in the language and acoustic models because the introduced ASR framework should react perfectly in different speech environments. Therefore, only the system ensures a higher recognition rate.

Consider that the extracted features are had at T-length, and the features are defined as $X = \{ X_T \in \mathbb{R}^D | t = 1, \ldots, T| \}$. The features are extracted for the length of the spoken word and defined as $W = \{ W_n \in v | n = 1, \ldots, N| \}$. The features $X$ are derived from $t$ frame and $W$ word position $n$ and $v$ vocabulary in D-dimension. The derived features are further examined to get the acoustic features that are obtained from the most likely appearing words:

$$\hat{W} = argmax_w P(W|X) \tag{15}$$

The acoustic feature $P(W|X)$ is computed from the word sequence $W$ from $X$ using Bayes' rules, defined in Equation (16). During the computation, $P(X)$ is omitted when the word is constant, belonging to the word $W$.

$$\hat{W} = argmax_w \frac{P(X|W).P(W)}{P(X)} \tag{16}$$

$$\hat{W} = argmax_w P(X|W).P(W) \tag{17}$$

Then, the sequence of features $P(X|W)$ is computed from the acoustic model and the priori knowledge of the word $P(W)$ is computed from the language model. The sequence of features, words, and the respective analysis is performed using Equation (18).

$$\left.\begin{array}{l} argmax_{w \in v^*} P(W|X) \\ = argmax_{w \in v^*} \sum_S P(X|S, W), \ P(S|W) \ P(W) \\ \approx argmax_{w \in v^*} \sum_S P(X|S), \ P(S|W) \ P(W) \end{array}\right\} \tag{18}$$

$P(X|S)$ is derived from the acoustic model, which helps make the Markov assumption concerning the probabilistic chain rules (Equation (19)).

$$\left.\begin{array}{l} P(X|S) = \prod_{t=1}^{T} P(x_t|x_1, x_2, x_3 \ldots \ldots x_{t-1}, S) \\ \approx \prod_{t=1}^{T} P(x_t|S_t) \propto \prod_{t=1}^{T} \frac{P(S_t|X_t)}{P(S_t)} \end{array}\right\} \tag{19}$$

The convolution network changed the $P(x_t|S_t)$ frame-wise likelihood function into the frame-wise posterior distribution $\frac{P(S_t|X_t)}{P(S_t)}$. The frame-wise analysis helps to resolve the decision-making issues and the system's performance is improved by considering the lexicon model $P(S|W)$. This lexicon model process is factorized according to the Markov assumption and probabilistic model.

$$\left.\begin{array}{l} P(S|W) = \prod_{t=1}^{T} P(s_t|s_1, s_2, s_3 \ldots \ldots s_{t-1}, W) \\ \approx \prod_{t=1}^{T} P(s_t|s_{t-1}, W) \end{array}\right\} \tag{20}$$

The extracted phoneme features and respective Markov probability value helps to identify the lexicon information from the speech. Finally, the language model $P(W)$ is computed using the Markov assumption and probabilistic chain rule for a word in speech.

$$\left.\begin{array}{l} P(W) = \prod_{n=1}^{N} P(w_n|w_1, w_2, w_3 \ldots \ldots w_{n-1}) \\ \approx \prod_{t=1}^{T} P(w_n|w_{n-m-1, \ldots \ldots} \ldots \ldots w_{m-1}) \end{array}\right\} \tag{21}$$

The Appendix A is explained the sparse encoder and model Fine-Tuning using Haris Hawk optimization.

## 4. Results and Discussion

### 4.1. Experiment Setup

The collected datasets are investigated, in which 80% of the dataset is utilized as training and 20% is used for testing purposes. This process is developed using MATLAB (MathWorks Inc., Natick, MA, USA) and the system uses the acoustic and language model to train the networks. Here, people's speech information is investigated in every word, phenomena, and fluctuation that helps to identify every speech in different environments. During the analysis, the Harris Hawk optimization process is utilized to update and fine-tune network parameters to reduce the maximum error-rate classification problem. Further, the system's robustness and reliability are maintained by extracting the valuable features in all signal sub-bands and wavelets. Due to the effective analysis of the speech signal spectrum, power and modulations were used to remove the modulations and deviations in the captured speech signal.

### 4.2. Objective Performance Evaluation

This section determines how the proposed HHSAE-ASR framework obtains the substantial results while working on the speech recognition process. The system effectiveness is evaluated using the error rate values because it is more relevant to the maximum error-rate classification problem. The resultant value of the HHSAE-ASR is compared with the existing research works such as [12,14,15,18,20]. These methods, described in more detail in Section 2, were selected because of their utilization of the optimization techniques and functions while analyzing the speech signal.

Table 1 illustrates the error rate analysis of the proposed HHSAE-ASR framework which is compared with the existing algorithms, such as the multiobjective evolutionary optimization algorithm [12], the deep convolution encoder and long short term recurrent neural networks [14], continual learning algorithms [15], enhancement parameter with a genetic algorithm [18], and MFCC and DTW [20]. Among these methods, the HHSAE-ASR algorithm attains minimum error values (MSE—1.11, RMSE—1.087, and VUV—1.01). The training process uses different features like the acoustic, lexicon, and language model with the speech signal. These features are more helpful in making decisions according to the probability value and chain rules.

**Table 1.** Training values Error rate in Speech Processing.

| Methods | Mean Square Error (MSE) | Root Mean Square Error (RMSE) | Voice/Unvoiced (VUV) Error |
|---|---|---|---|
| Multiobjective evolutionary optimization algorithm [12] | 1.65 | 1.43 | 1.28 |
| Deep convolution encoder and LSTM-RNN [14] | 1.53 | 1.38 | 1.26 |
| Continual learning algorithms [15] | 1.427 | 1.25 | 1.17 |
| Genetic algorithm [18] | 1.36 | 1.14 | 1.15 |
| MFCC and DTW [20] | 1.21 | 1.12 | 1.10 |
| HHSAE-ASR | 1.11 | 1.087 | 1.01 |

Here, the set of speech features are analyzed by applying the encoder network that uses the different conditions while updating the network parameters,.The error rate has been evaluated on different numbers of users and the obtained results are illustrated in Figure 4.

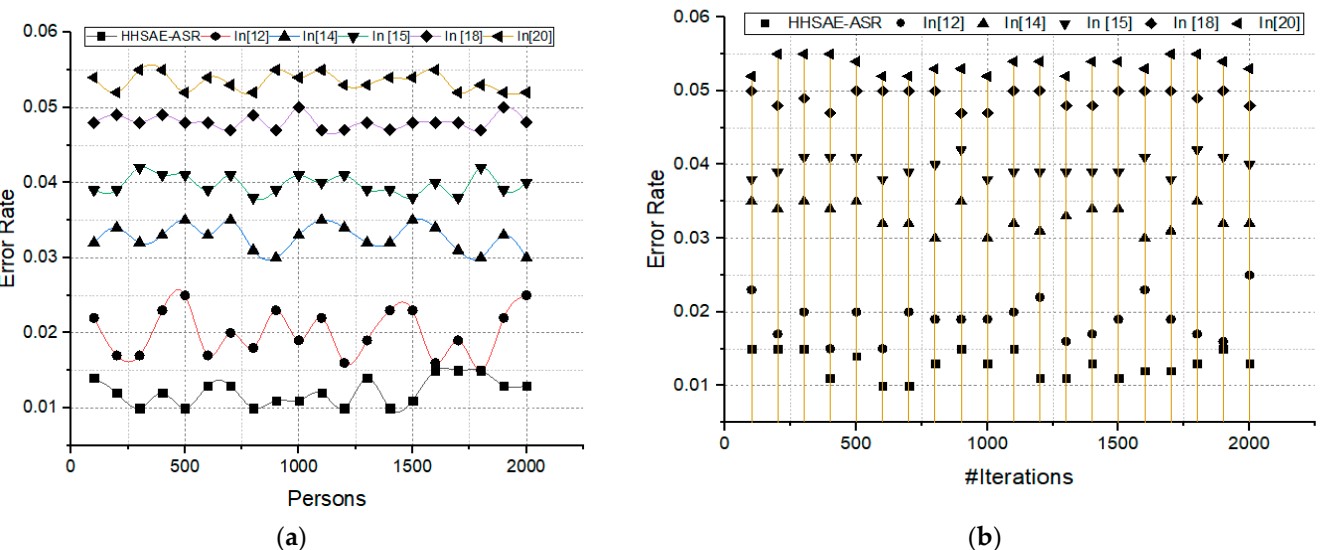

(**a**)　　　　　　　　　　　　　　(**b**)

**Figure 4.** Error rate analysis of various numbers of persons and iterations; (**a**) The comparisons of persons and error rate; (**b**) the iterations number and error rate plot.

Figure 4 illustrates the error rate analysis of the different number of persons that participated during the speech analysis process. The effective utilization of the speech features and training parameters helps to reduce the classification error rate. The minimum error rate directly indicates the maximum recognition accuracy on the objective analysis. The obtained results are illustrated in Figure 5.

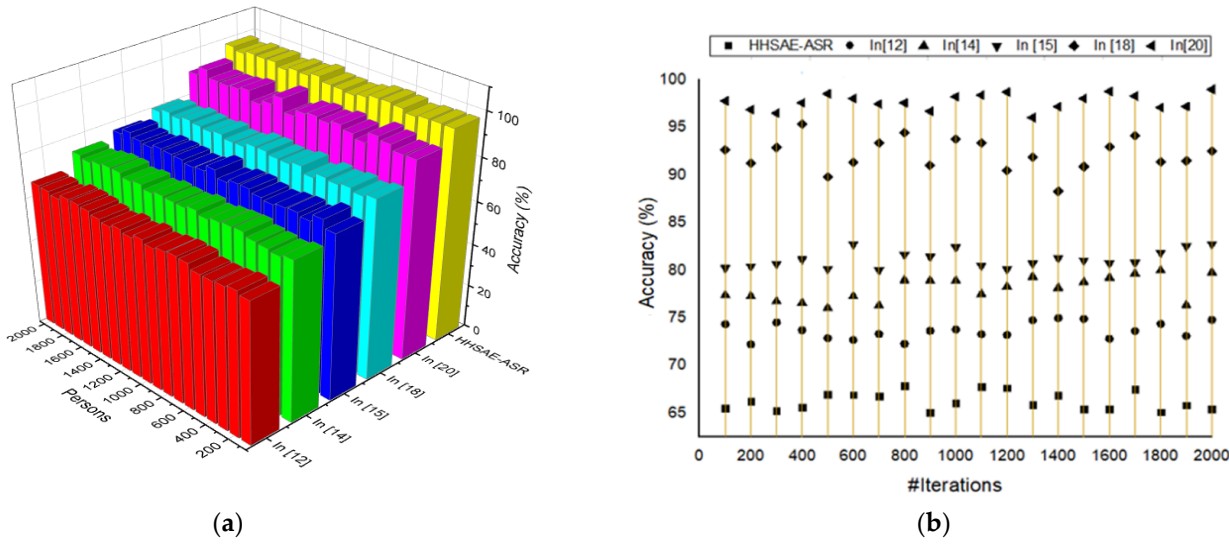

(**a**)          (**b**)

**Figure 5.** Training accuracy; (**a**) Training accuracy vs number of persons; (**b**) accuracy against number of iterations.

The above results illustrate that the proposed HHSAE-ASR framework attains effective results while investigating the speech signals on a different number of iterations and persons. The recognition system's effectiveness is further examined using the testing model for a different number of persons and iterations in the subjective analysis.

### 4.3. Subjective Performance Evaluation

This section discusses the performance evaluation results of the HHSAE-ASR framework in a subjective manner. The dataset consists of much recorded information that is both male and female. Therefore, the testing accuracy is determined using various numbers of persons and iterations.

Figure 6 shows that the proposed HHSAE-ASR framework attains high accuracy (98.87%) while analyzing various people's signals on a different number of iterations. The obtained results are compared to existing methods: multiobjective evolutionary optimization algorithm [12] (66.76%), deep convolution encoder and long short term recurrent neural networks [14] (73.43%), continual learning algorithms [15] (78.31%), enhancement parameter with a genetic algorithm [18] (81.34%), and MFCC and DTW [20] (93.23%).

Table 2 illustrates the excellency of the introduced system's efficiency while investigating a different number of participants. The system examined each person's speech signal as it compared the speech word, length, and sequence-related probability value. The Markov chain rules developed according to the acoustic model, lexicon model, and language model, which helps to identify the speech relationships and their deviations in the loud and noisy environment.

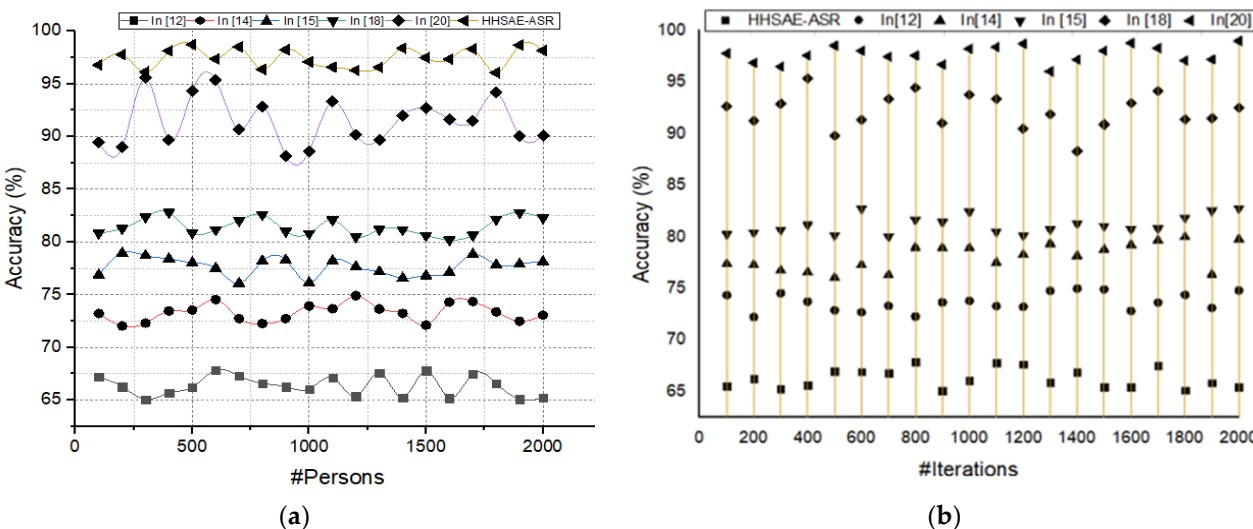

**Figure 6.** Testing accuracy; (**a**) Test accuracy vs number of persons; (**b**) test accuracy vs the number of iterations.

**Table 2.** Results of subjective evaluation.

| Participants | Precision (%) | Recall (%) | Mathew Correlation Coefficient (MCC) (%) | F-Measure (%) |
|---|---|---|---|---|
| 100 | 99.53 | 99.02 | 99.21 | 99.24 |
| 150 | 99.24 | 99.35 | 99.5 | 99.35 |
| 200 | 98.56 | 99.21 | 99.23 | 99.23 |
| 250 | 99.13 | 99.46 | 99.1 | 99.56 |
| 300 | 99.56 | 99.45 | 99.43 | 99.22 |
| 350 | 99.25 | 99.13 | 99.00 | 99.26 |
| 400 | 99.22 | 99.54 | 99.23 | 99.23 |
| 450 | 99.21 | 99.24 | 99.18 | 99.35 |
| 500 | 99.56 | 99.56 | 99.34 | 99.03 |
| 550 | 99.13 | 99.12 | 99.39 | 99.13 |
| 600 | 99.11 | 99.02 | 99.12 | 99.23 |
| 650 | 99.09 | 99.23 | 99.02 | 99.3 |
| 700 | 99.15 | 99.76 | 98.92 | 99.22 |
| 750 | 99.25 | 98.37 | 99.032 | 98.77 |
| 800 | 99.35 | 99.02 | 99.21 | 98.3 |
| 850 | 99.53 | 98.98 | 99.34 | 99.45 |
| 900 | 99.21 | 99.12 | 99.10 | 99.56 |
| 950 | 99.02 | 99.3 | 99.3 | 99.1 |
| 1000 | 99.13 | 99.33 | 99.42 | 98.98 |

　　　　Thus, the proposed HHSAE-ASR system recognizes the speech synthesis with 99.31% precision, 99.22% recall, 99.21% of MCC, and 99.18% of F-measure value.

　　　　Table 2 illustrated the excellence of the introduced system's efficiency while investigating a different number of participants. It analyzes each person's speech signal based on their word length, sequence-related probability, and the chain rules that are taken between 100 to 1000 participants. The method predicts the sequence of features $P(X|W)$ and respective $argmax_{w \in v^*} \sum_S P(X|S)$, $P(S|W)P(W)$ values help to match the training and testing features.

　　　　Further, the method updates the network parameters in different conditions $|E| \geq 1$ which consider the rabbit's every movement, lower and upper boundary $X_{rand}(t) - r_1|X_{rand}(t) - 2r_2X(t)|$; $q \geq 0.5$, and $(X_{rabbit}(t) - X_m(t)) - r_3(LB + r_4(UB - LB))$; $q < 0.5$ conditions. Not only this, but the system uses the $|E| < 1(r \geq 0.5 \ and \ |E| \geq 0.5)$ and $|E| < 1(r < 0.5 \ and \ |E| \geq 0.5)$.

The new system's efficiency improves when tested with various participants. It analyzes each person's speech signal based on their word length, sequence-related probability, and chain rules. The approach predicts the sequence of features and their respective values, which helps to match the training and testing features.

In the HHSAE-ASR framework, speech patterns are continuously used to train the system. The encoder network is then fine-tuned using metaheuristic techniques to reduce the error rate classification problem to a minimum. The ASR accuracy, robustness, and dependability are enhanced by using sequence speech patterns, learning concepts, and network parameter updating.

### 4.4. Data Accessing in HHSAE-ASR

The recognition and authentication of human speech uses dynamic time wrapping (DTW). These techniques are used to extract the distinctive aspects of human speech. It is easier to authenticate users using the derived features. Thus, this system's total security and authentication efficiency is enhanced with an achievement of 91.8%. The accessing of data in our proposed system is compared with other traditional approaches that are given in Table 3.

**Table 3.** Data accessing analysis.

| Number of Participants | MFCC (%) | MSE (%) | HHSAE-ASR (%) |
|:---:|:---:|:---:|:---:|
| 10 | 47.2 | 50.4 | 71.4 |
| 20 | 48.4 | 53.6 | 74.7 |
| 30 | 49.5 | 56.8 | 76.3 |
| 40 | 52.7 | 58.5 | 77.7 |
| 50 | 54.8 | 60.7 | 81.9 |
| 60 | 55.9 | 62.2 | 83.1 |
| 70 | 59.4 | 61.3 | 85.2 |
| 80 | 60.6 | 65.6 | 88.5 |
| 90 | 63.8 | 68.8 | 90.6 |
| 100 | 66.2 | 69.2 | 91.8 |

This kind of validation helps to reduce the classification error rate compared to other methods. Thus, the Harris Hawks sparse auto-encoder networks (HHSAE-ASR) system recognizes the speech synthesis with 99.31% precision, 99.22% recall, 99.21% MCC, and 99.18% F-measure value.

## 5. Conclusions

This paper proposed the Harris Hawks sparse auto-encoder network (HHSAE-ASR) framework for automatic speech recognition. Initially, the human voice signal is collected and analyzed by using the spectrum decomposition approach. Here, spectrum deviations and fluctuations are analyzed to replace the noise signal with the average spectrum phase value. Then, different features are extracted from the signal by decomposing the signals into four levels. The decomposed signals are further investigated to get the Mel-frequency coefficient features, which are more useful to create the acoustic, lexicon, and language model. The extracted features are applied to the Markov model-based convolution network to train the network for resolving the loud and noisy environment speech signal analysis. During this process, the network is fine-tuned, and the parameters are updated according to the Harris hawk prey searching behavior with certain updating conditions. This process reduces misclassification error rate problems and maintains the robustness and availability of the system. Thus, the system ensures a 99.18% accuracy, which outperforms the existing algorithms.

Natural language recognition is a challenging task, as different dialects, speeds, and traditions vary in actual applications. In the future, a relevant feature selection process will be incorporated to improve the overall effectiveness of the system. By using Mel-frequency cepstral coefficients to express the characteristics, the correctness of the classification could

improve. It will be useful to integrate deep learning algorithms into the classifier design instead of traditional methods.

**Author Contributions:** Conceptualization, A.R.; Data curation, M.H.A.; Formal analysis, M.M.J., S.K.A., M.J.A., D.V.-A., R.D. and S.A.B.; Funding acquisition, D.V.-A.; Investigation, M.H.A., M.M.J. and S.K.A.; Methodology, A.R.; Software, M.H.A.; Validation, S.K.A., A.R., M.J.A., R.D. and S.A.B.; Writing—original draft, M.H.A., M.M.J., S.K.A., A.R., M.J.A. and S.A.B.; Writing—review & editing, D.V.-A. and R.D. All authors have read and agreed to the published version of the manuscript.

**Funding:** This research received no external funding.

**Institutional Review Board Statement:** Not applicable.

**Informed Consent Statement:** Not applicable.

**Data Availability Statement:** Datasets are available at https://github.com/jim-schwoebel/voice_datasets (accessed on 18 April 2019).

**Conflicts of Interest:** The authors declare no conflict of interest.

## Appendix A

*Appendix A.1 Sparse Autoencoding*

According to the acoustic, lexicon, and language model above, the speech features are trained in the convolution network to reduce the maximum error-rate classification issues. This learning presentation is further improved by applying the sparse auto encoder (SAE), which improves the classification performance. This computation uses the maximum hidden units compared to the inputs but allows only a small number of hidden units to compute the output. The network uses acoustic, language, and other features extracted to perform the speech recognition process differently. The training process $L(x, x') + \Omega(h)$ uses the sparsity penalty $\Omega(h)$ for the code layer $h$, and the output is achieved as $h = f(Wx + b)$. Here, the penalty helps compute the output; if the model belongs to a particular input model, then the penalty encourages the output as one, else 0; this penalty process is achieved via the Kullback–Leibler divergence in Equation (A1).

$$\hat{\rho}_j = \frac{1}{m} \sum_{i=1}^{m} \left[ h_j(x_i) \right] \tag{A1}$$

For training sample $m$, hidden unit $j$, the average activation function, is applied to compute the output value in the hidden layer $h_j(x_i)$ of input($x_i$). Suppose the calculated value is inactive to the input, then the $\hat{\rho}_j$ value is 0. Hence, the $\hat{\rho}_j$ value must be the sparsity parameter $\rho$. Then, the KL divergence of the sparsity parameter is computed in Equation (A2).

$$\sum_{j=1}^{s} KL(\rho || \hat{\rho}_j) = \sum_{j=1}^{s} \left[ \rho log \frac{\rho}{\hat{\rho}_j} + (1 - \rho) log \frac{1 - \rho}{1 - \hat{\rho}_j} \right] \tag{A2}$$

*Appendix A.2 Model Fine-Tuning using Haris Hawk Optimisation*

The network has to be fine-tuned to improve the ASR system's performance. The network parameters are adjusted for reducing the error-rate classification problem. To achieve this goal, the parameters are analyzed in the N population size with an extreme number of iterations T. The output network parameters are identified according to the rabbit food searching characteristics and the objective function.

In the population, $(X_i; i = 1, 2, \ldots, N)$, the fitness value of hawks is computed with respect to the rabbit's position. Initially, the hawk's position $X_{ra}$ is the best location. For every iteration, the rabbit's initial energy $(E_0)$ is checked along with the jump strength $(j)$, defined in Equation (A3).

$$\left. \begin{array}{c} E_0 = 2\, rand() \\ J = 2(1 - rand()) \end{array} \right\} \tag{A3}$$

This computed energy and jump strength value is updated for every jump and food searching process using Equation (A4), as it is used to identify the best network parameter value.

$$E = 2E_0 \left( 1 - \frac{t}{T} \right) \tag{A4}$$

The energy value is updated according to the prey energy value while escaping (E) on the maximum iteration T with initial energy $E_0$. The $E_0$ value is selected between $(-1, 1)$, which determines the hawk's condition. If the value $E_0$ is reduced between 0 to $-1$. If $|E| \geq 1$ (exploration phase), then it moves to a different location, and it updates continuously for selecting the effective network parameter. If $|E| < 1$, then the rabbit is in the neighborhood phase searching for the solution in the exploitation step. As said, if $|E| \geq 1$, it is in the exploration phase; then, the location vector is updated using Equation (A5).

$$X(t+1) = \begin{cases} X_{rand}(t) - r_1 |X_{rand}(t) - 2r_2 X(t)| \; q \geq 0.5 \\ (X_{rabbit}(t) - X_m(t)) - r_3(LB + r_4(UB - LB)) \; q < 0.5 \end{cases} \tag{A5}$$

The updating of the $|E| \geq 1$ condition is the next iteration of the hawk's position $X(t+1)$ updating process that is done by the rabbit position $X_{rabbit}(t)$, the hawk's current position vector $X(t)$, and the random numbers $r_1, r_2, r_3, r_4$ and q having values of (0, 1). For every iteration, the lower (LB) and upper (UB) boundary of the searching region is considered with the current population $X_{rand}(t)$ and the position $X_m$ of hawks. Suppose $|E| < 1$ ($r \geq 0.5 \; and \; |E| \geq 0.5$), then it goes to the exploitation phase, and the energy factor is updated using Equation (A6).

$$\left. \begin{array}{l} X(t+1) = \Delta X(t) - E |JX_{rabbit}(t) - X(t)| \\ \Delta X(t) = X_{rabbit(t)} - X(t) \end{array} \right\} \tag{A6}$$

The updating process is performed by computing the difference between the location and the position vector of the rabbit in every iteration $t$. Here, the jumping strategy $J$ is estimated as $J = 2(1 - r5)$; the random number is computed between (0, 1). The jumping value is changed in every iteration because the rabbit moves in the search space randomly. Suppose the $|E| < 1$ ($r \geq 0.5 \; and \; |E| < 0.5$), then the updating process is performed as:

$$X(t+1) = X_{rabbit}(t) - E \, |\Delta X(t)| \tag{A7}$$

This updating process is performed when the Harris hawk has a low escaping energy level; then, the updating of the current position is done as Equation (A7). If $|E| < 1$ ($r < 0.5 \; and \; |E| \geq 0.5$), then the location vector is updated using Equation (A8).

$$X(t+1) = \begin{cases} Y \; if \; F(Y) < F(X(t)) \\ Z \; if \; F(Z) < F(X(t)) \end{cases} \tag{A8}$$

Here, $Y$ and $Z$ are computed as follows,

$$Y = X_{rabbit}(t) - E |JX_{rabbit}(t) - X(t)| \tag{A9}$$

$$Z = Y + S * LF(D) \tag{A10}$$

Here, $Y$ and $Z$ parameters are computed in the $D$ dimension, the Levy flight function $LF$, and the random vector ($S$) with $D$ size. The $LF$ is computed as follows,

$$\left. \begin{array}{l} LF(x) = 0.01 * \dfrac{u * \sigma}{|v|^{\frac{1}{\beta}}} \\[2mm] \sigma = \left( \dfrac{\Gamma(1+\beta) * sin\left( \frac{\pi\beta}{2} \right)}{\Gamma\left( \frac{1+\beta}{2} \right) * \beta * 2^{\left( \frac{\beta-1}{2} \right)}} \right)^{1/\beta} \end{array} \right\} \tag{A11}$$

Here, random values between (0, 1) are selected for $u$ and $v$ and 1.5 is the constant value for $\beta$. At last, the $|E| < 1$ ($r < 0.5$ *and* $|E| < 0.5$), then the updating process is done by using Equation (A12).

$$X(t+1) = \begin{cases} Y \; if \; F(Y) < F(X(t)) \\ Z \; if \; F(Z) < F(X(t)) \end{cases} \tag{A12}$$

Here, $Y$ and $Z$ are computed as follows,

$$Y = X_{rabbit}(t) - E|JX_{rabbit}(t) - X_m(t)| \tag{A13}$$

$$Z = Y + S * LF(D) \tag{A14}$$

$$X_m(t) = \frac{1}{N}\sum_{i=1}^{N} X_i(t) \tag{A15}$$

According to this process, the network parameters are updated continuously, which reduces the recognition issues and the existing research problem. Based on the encoder network performance, the convolute network identifies the speech by examining the acoustic, lexicon, and language model effectively.

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
