# Peer review of "Harris Hawks Sparse Auto-Encoder Networks for Automatic Speech Recognition System"

_applsci, doi:10.3390/app12031091_

Round 1

Reviewer 1 Report

1) Readability is poor. Writing quality must be improved otherwise difficult to understand the meaning. e.g "the developed ASR system has time." (Line56) & "allow  users to speak the machine..." (Line50) & "The recognition system must provide for variations of voices" (Line 57) & "The system examining each person .... to identify the  speech relationship and their deviations in the loud and noisy environment. (Line 500-504)

2) Figure quality should be improved. e.g. Figure 7 (a) triangles with different directions are hard to read.

3)  Subjective evaluation on HHSAE-ASR  system (Line505-506) can't match results listed in Table 2.

4) One contribution described by authors is “maintain reliability, flexibility, and robustness of the system". However, there is a lack of solid experimental results to support the claims. For example, algorithm performance under different signal-noise-rate conditions should be verified in order to evaluate the robustness as well as accents and speaking speeds. In Section 4, redundant results are listed or plotted (e.g. Table 1 and Figure 4).

5) Equations (e.g. (4) & (5)) must provide detailed explanation instead of mathematical symbols. Reference note may be a better solution for complicated expression.

Author Response

Response to Reviewer 1 Comments

Dear Reviewer 1,

We would like to thank you for reviewing our paper. Your feedback has greatly contributed towards the improvement of this paper. All your comments have been thoroughly considered and the required changes have been made. The paragraphs below represent the step-by-step comments that you have made, and our response to them.

Point 1: Readability is poor. Writing quality must be improved otherwise difficult to understand the meaning. E.g. “the developed ASR system has time. “(Line56) &” allow users to speak the machine…” (Line 50) & “The recognition system must provide for variations of voices” (Line 57) & “The system examining each person… to identify the speech relationship and their deviations in the loud and noisy environment (Line 500-504).

Response 1:  As per the able reviewer valuable comments, the above statements are updated and verified and highlighted .

 Point 2: Figure quality should be improved. E.g. Figure 7(a) triangles with different directions are hard to read.

Response 2: Figure resolution is improved in the manuscript with high resolution also changed in Figure 5(a).

 Point 3: Subjective evaluation on HHSAE-ASR system (Line 505-506) can’t match results listed in table 2.

Response 3: The evaluation on HHSAE-ASR in updated manuscript. For this the lines are added

[Line # 447 to 455]. Table 3 is added about data accessing analysis

 Point 4: One contribution described by authors is “maintain reliability, flexibility, and robustness of the system”. However, there is a lack of solid experimental results to support the claims. For example, algorithm performance under different signal-noise-rate conditions should be verified in order to evaluate the robustness as well as accents and speaking speeds. In section 4, redundant results are listed or plotted (e.g. table 1 and figure 4).

Response 4: Yes, able reviewer is correct, there is one main contribution of our study. There is no contribution regarding maintain reliability, flexibility and robustness of the system. In the updated manuscript we removed this sentence in the introduction section [Line # 83-84].

That our main contribution of the proposed system is reducing the maximum error rate problem and improving the precision of the ASR system.

 The figure 4 is removed from the manuscript.

Point 5: Equations (e.g. (4) &(5)) must provide detailed explanation instead of mathematical symbols. Reference note may be a better solution for complicated expression.

Response 5:  As per the able reviewer the equations 4 and 5 are provided the explanation [Line # 256- 277].

Reviewer 2 Report

This paper demonstrates that the network performance can be improved by applying Harris Hawks Optimization so it can effectively recognize speech in a noisy environment. I found the paper quite interesting. However, it should be improved to have a relevant scientific soundness and a better structure.

The results section is not discussed at all and it only takes 2 pages out of 17. I recommend authors to discuss the results and extend the results section. I highly recommend authors to include a new Discussion section.

Table 2. Results are quite similar from 100 to 1000 participants. Can you discuss why? How do you obtain 99.31% precision? 

Methods section is quite extensive (5-6 pages) and hard to follow. I suggest authors move all mathematical formulation and ASR fundamentals to an Appendix. Also please explain better at the beginning of each section the contents that are included.

I also miss the definition of the following particular words: fitness, rabbit, prey.

Kaldi-ASR is mentioned but not introduced before.

Literature review. I highly recommend authors to include references about the use of ASR in computer-assisted pronunciation training due to the latest improvement in ASR technology under different scenarios: https://doi.org/10.1109/ACCESS.2020.2988406, https://dr.lib.iastate.edu/handle/20.500.12876/23583

English language and style are fine/minor spell check required.

Author Response

Response to Reviewer 2 Comments

Dear Reviewer 2,

We would like to thank you for reviewing our paper. As you commented that paper demonstrates that the network performance can be improved by applying Harris Hawks Optimization so it can effectively recognize speech in a noisy environment. I found the paper quite interesting. Your feedback has contributed to the betterment of this paper. All your comments have been considered and the required changes have been made. The paragraphs below represent the step-by-step comments that you have made, and our response to them.

Point 1: The results section is not discussed at all and it only takes 2 pages out of 17. I recommend authors to discuss the results and extend the results section. I highly recommend authors to include a new Discussion section.

Response 1As per the able reviewer suggestion, the result and discussion section are extended.

The new section 4.3 is added Data accessing in HHSAE-ASR.

Table 3 is added in the manuscript.

[Line # 433 to 469].

Point 2: Table 2. Results are quite similar from 100 to 1000 participants. Can you discuss why? How do you obtain 99.31% precision?

Response 2: The effectiveness of the new system's efficiency improves when tested with a variety of participants. It analyses each person's speech signal based on their word length, sequence-related probability, and chain rules are taken between 100 to 1000 participants. The approach predicts the sequence of features and their respective values, which helps to match the training and testing features.

Point 3: Methods section is quite extensive (5-6 pages) and hard to follow. I suggest authors move all mathematical formulation and ASR fundamentals to an Appendix. Also please explain better at the beginning of each section the contents that are included.

Response 3: The section 3.5. Sparse auto encoding mathematical formulations are moved to under the heading Appendix at the end of the paper.

Point 4: I also miss the definition of the following particular words: fitness, rabbit, prey.

Response 4: According to the rabbit's traits and goal, the output network parameters are determined. The fitness value of hawks in a population should be calculated in relation to the location of the rabbit. At first glance, the Hawks' position is the best one. The initial energy of the rabbit and its jumping strength should be evaluated for each iteration. An energy factor is used to update the hawk's parameters during its exploration and exploitation of prey. Every time a rabbit makes a move, an update is made to the location.

Point 5: Kaldi-ASR is mentioned but not introduced before.

Response 5:  The Kaldi-ASR is mentioned in the literature with introduction

[Line # 105- 109]

Point 6: Literature review. I highly recommend authors to include references about the use of ASR in computer-assisted pronunciation training due to the latest improvement in ASR technology under different scenarios: https://doi.org/10.1109/ACCESS.2020.2988406, https://dr.lib.iastate.edu/handle/20.500.12876/23583

Response 6:  Able reviewer is correct these references are important for our research for ASR in computer-assisted pronunciation.

Added as refs. [45, 46] in the literature review.

Point 7: English language and style are fine/minor spell check required.

Response 7: The spell check is verified and updated in the manuscript through the use of appropriate tools (grammarly) and with the help of a native speaker.

Round 2

Reviewer 1 Report

Natural language recognition is a challenge task as different dialects, speed and traditions are varying in actual applications. By using Mel-frequency cepstral coefficients to express the characteristics, correctness of classification is improved. It will be useful to integrate deep learning algorithms into the classifier design instead of traditional methods (e.g. HMM)

Author Response

Dear Reviewer 1,

We would like to thank you again for reviewing our paper. Your feedback has contributed to the betterment of this paper. All your comments have been considered and the required changes have been made.

Point 1: Natural language recognition is a challenge task as different dialects, speed and traditions are varying in actual applications. By using Mel-frequency cepstral coefficients to express the characteristics, correctness of classification is improved. It will be useful to integrate deep learning algorithms into the classifier design instead of traditional methods (e.g. HMM)

Response 1:  As per the able reviewer valuable comments these points are added in our future work.          [Line # 487-492]

Reviewer 2 Report

Authors change have improved the overall quality of the manuscript. Appendix section shoudl be after the references.

Author Response

Dear Reviewer 2,

We would like to thank you again for reviewing our paper.

Point 1: Authors change have improved the overall quality of the manuscript. Appendix section should be after the references.

Response 1As per the able reviewer suggestion, the appendix section is included after references. ________________________________________________________________________